# Impact of Food-Based Weight Loss Interventions on Gut Microbiome in Individuals with Obesity: A Systematic Review

**DOI:** 10.3390/nu14091953

**Published:** 2022-05-06

**Authors:** Aleisha Bliesner, Jade Eccles-Smith, Claire Bates, Olivia Hayes, Jet Yee Ho, Catia Martins, Helen Truby, Marloes Dekker Nitert

**Affiliations:** 1School of Human Movement and Nutrition Sciences, The University of Queensland, Brisbane, QLD 4067, Australia; a.bliesner@uq.edu.au (A.B.); c.bates@uq.net.au (C.B.); o.hayes@uq.net.au (O.H.); jetyee.ho@uq.net.au (J.Y.H.); h.truby@uq.edu.au (H.T.); 2Department of Obstetric Medicine, The Royal Brisbane and Women’s Hospital, Brisbane, QLD 4029, Australia; jade.ecclessmith@uqconnect.edu.au; 3Mater Research Institute, The University of Queensland, Brisbane, QLD 4101, Australia; 4Department of Clinical and Molecular Medicine, Norwegian University of Science and Technology, 7034 Trondheim, Norway; catia197@uab.edu; 5Centre for Obesity and Innovation (ObeCe), St. Olav University Hospital, 7006 Trondheim, Norway; 6Department of Nutrition Sciences, University of Alabama at Birmingham, Birmingham, AL 35294, USA; 7School of Chemistry and Molecular Biosciences, The University of Queensland, Brisbane, QLD 4067, Australia

**Keywords:** diet, weight loss, obesity, microbiota, microbiome, alpha-diversity, beta-diversity, short-chain fatty acids

## Abstract

The observation that the gut microbiota is different in healthy weight as compared with the obese state has sparked interest in the possible modulation of the microbiota in response to weight change. This systematic review investigates the effect of food-based weight loss diets on microbiota outcomes (α-diversity, β-diversity, relative bacterial abundance, and faecal short-chain fatty acids, SCFAs) in individuals without medical comorbidities who have successfully lost weight. Nineteen studies were included using the keywords ‘obesity’, ‘weight loss’, ‘microbiota’, and related terms. Across all 28 diet intervention arms, there were minimal changes in α- and β-diversity and faecal SCFA concentrations following weight loss. Changes in relative bacterial abundance at the phylum and genus level were inconsistent across studies. Further research with larger sample sizes, detailed dietary reporting, and consistent microbiota analysis techniques are needed to further our understanding of the effect of diet-induced weight loss on the gut microbiota.

## 1. Introduction

The rate of overweight and obesity is steadily increasing worldwide, affecting over 1.9 billion adults in 2016 according to the World Health Organization [1]. Obesity is associated with a number of chronic diseases including cardiovascular disease, type 2 diabetes, and certain cancers, which place a significant economic burden on the healthcare system [2,3]. Whilst the obese state has traditionally been attributed to energy intake in excess of energy expenditure, more recently, genetics, epigenetics, and the microbiome have been implicated in the aetiology of obesity [4,5]. Lifestyle modification, including dietary changes, remains the recommended first-line intervention for weight loss.

The human colon is home to 10^14^ microorganisms, which interact with multiple systems to influence host health [6]. Microbial composition and short-chain fatty acid (SCFA) production are both influenced by the host diet and the composition of the microbiota has been implicated in the development and maintenance of the obese state. Ley et al. [7] first observed that individuals with obesity had a higher ratio of Firmicutes to Bacteroidetes compared to their lean counterparts. Ridaura et al. [8] demonstrated a causal relationship between the microbiota and obesity by transplanting faecal microbiota from twins discordant for obesity into germ-free mice. Mice receiving faecal transplants from the obese twins gained significantly more weight than those transplanted with the microbiota of the lean twins, and this correlated with lower SCFA production in the obese recipient mice.

Numerous studies have aimed to identify the most successful macronutrient composition for weight loss. A recent meta-analysis has shown both low-fat and low-carbohydrate diets to be effective for weight loss, with little difference between the two [9]. Long-term weight maintenance, however, remains difficult, with only 28% of adults maintaining a loss of 10% body weight after 4 years [10]. While similarly effective for short-term weight loss, different macronutrient ratios may affect long-term success, potentially via modulation of the gut microbiota, though the exact mechanism remains unknown.

The purpose of this systematic review was to investigate the effect of food-based weight loss diets varying in macronutrient composition on microbiota outcomes (α- and β-diversity, relative bacterial abundance, and SCFA production) in healthy but obese individuals who had lost at least 2 kg of weight attributed to changing their food intake.

## 2. Materials and Methods

### 2.1. Search Strategy

An electronic literature search was performed on the following databases: PubMed, Scopus, CINAHL, and Embase. Searches were performed on the same day, without filters, including literature from database inception until 26 October 2021. Database searches were performed using the terms ‘overweight’ or ‘obesity’ and ‘weight loss’, and ‘microbiota’ or ‘microbiome’ and related terms (see Appendix A).

### 2.2. Study Selection

Studies that met the following criteria were included: (1) subjects must be healthy adults (18–70 years old) with overweight or obesity (BMI > 25 kg/m^2^), (2) subjects must have achieved ≥2 kg weight loss induced by a hypocaloric diet or a combination of a hypocaloric diet and other lifestyle interventions, and (3) studies that assessed α-diversity, β-diversity, bacterial abundance, or SCFA concentrations. Studies were excluded if they were: (1) published in a language other than English, (2) an abstract only, (3) conducted in animals, children, adolescents, pregnant women, or subjects with chronic illnesses/morbidities, (4) used pre- or probiotics only, faecal microbiota transplant, herbal medicines, pharmacotherapy, a single food only (e.g., avocado), Glucagon-like-peptide 1 (GLP-1) or other gut hormones/peptides, or supplement-based very-low-calorie diets (e.g., Optifast). If a multi-arm trial had a single weight loss intervention arm meeting the above criteria, this specific arm was included and treated as a single-arm trial for analysis.

References were imported into an online screening and data-extraction tool (Covidence, v2815) and duplicates were removed following a 2-step process: automatic removal by Covidence, followed by manual removal by two reviewers. The two independent reviewers (A.B. and J.E.-S.) screened articles based on title and abstract against the eligibility criteria. Studies included were assessed based on their full text to produce the final selection of eligible studies. Disagreements were resolved through consensus-based discussions or by a third reviewer’s opinion.

### 2.3. Data Extraction

Two independent reviewers (A.B. and J.E.-S.) extracted data from each full-text including Appendix A using a pre-specified data-extraction template. Information on the first author, publication year, country where the study was conducted, study design, type of intervention, sample size, participant characteristics (sex, age, BMI), duration of intervention, and changes in weight, α- and β-diversity, bacterial abundance, and SCFAs were extracted. Disagreements were resolved through consensus or with a third reviewer. We did not contact study authors for additional information.

### 2.4. Risk of Bias

A risk-of-bias (RoB) assessment was conducted for each study by two independent reviewers (A.B. and J.E.-S.) using the Cochrane RoB tool [11]. The older version of the tool was used as the updated RoB2 is less suited to dietary studies due to difficulties with blinding and a lack of placebo in dietary interventions. RoB was assessed on the basis of sequence generation, allocation concealment, the blinding of participants and personnel for all outcomes, incomplete outcome data for faecal microbiota composition, selective outcome reporting, and other sources of bias. Each criterion was graded as having a high, low, or unclear RoB. Discrepant assessments were resolved by consensus reached through discussion between the two reviewers. 

## 3. Results

### 3.1. Study Selection

A total of 2741 records were retrieved by the database search, out of which 1290 duplicates were removed. A further 1367 records were excluded after title/abstract screening and the full-texts of the remaining 84 articles were assessed for eligibility. Sixty-five articles were excluded following full-text review while nineteen studies were retained for inclusion in this review. A detailed flowchart showing the study selection process is presented in Figure 1. Excluded full-texts with justifications are provided in Appendix A.

### 3.2. General Study Characteristics

Characteristics of the included studies are presented in Table 1. A total of 28 dietary interventions of interest were identified across the 19 included studies. Geographically, seven (37%) of the studies were performed in North America, seven (37%) in Europe, three (16%) in the United Kingdom, and two (11%) in East Asia. The publication dates of the included studies ranged from 2006 to 2021, with 16 (84%) studies published within the last 3 years. The mean number of participants enrolled in each dietary intervention arm was 39 (range, 6–97). Two studies (11%) were carried out exclusively in men, fifteen (79%) included both men and women, and two (11%) did not report on the sex of the participants. Mean age of participants ranged from 37 to 64 years old. Participants were either overweight or obese at baseline, with mean BMIs ranging from 26.6 to 36.6 kg/m^2^. Duration of each dietary intervention arm ranged from 10 days to 12 months. Fourteen studies used 16S rRNA gene amplicon sequencing to characterise the gut microbiota, two studies used shotgun metagenomic sequencing, one study used 16S rRNA-based quantitative FISH, and one study used a combination of qPCR and 16S rRNA amplicon sequencing.

### 3.3. Dietary Intervention Characteristics

Fourteen (74%) studies reported on the macronutrient intake of participants during the intervention (Table 2 and Table 3). Macronutrient distribution ranged from 12% to 34% of energy from protein, 16% to 73% of energy from carbohydrates, and 13% to 50% of energy from fat. Energy intake, reported by 11 studies (58%), ranged from 1195 to 2154 kcal/day. Eleven studies (58%) reported on dietary fibre intake, which ranged from 10 to 33 g per day, and three (16%) reported on the amount of soluble and insoluble fibre consumed.

### 3.4. Weight Loss

The mean weight loss across the 28 interventions was ~6 kg (Table 2 and Table 3). The lowest amount of weight loss achieved was 2.8 kg in two interventions lasting 3 and 8 weeks, respectively [16,22], while the largest amount of weight lost was 15.4 kg in a year-long intervention [7].

### 3.5. Changes in α-Diversity

Twenty-four interventions (86%) reported on α-diversity changes (Table 2). Eighteen (75%) produced no changes in α-diversity, while three (13%) increased α-diversity. These three interventions included an 8-week high-protein diet [16], a 6-month Mediterranean diet [20], and a 12-week energy-restricted diet [28]. A 16-week low-fat diet increased α-diversity in men but not women [14], while another study found an increase in OTU richness but not Shannon (diversity and richness) index following a 12-week weight reduction program [15]. One study which assigned omnivores to a 16-week vegan diet resulted in a decrease in α-diversity [24], however no changes were found following a similar intervention from the same research group [23].

### 3.6. Changes in β-Diversity

While not directly related to health outcomes, changes in β-diversity (i.e., the interindividual variation in microbiome composition) indicate whether an intervention had an overall effect on the microbiota. Seventeen interventions (61%) included in this review reported on β-diversity, eleven (65%) of which found no changes post-intervention (Table 2). Two (12%) resulted in a decrease in β-diversity: a 4-week low-carbohydrate, high-fat diet [21] and a 12-week low-carbohydrate diet [25]. β-diversity decreased at 3 months on another low-carbohydrate diet, but returned to baseline levels by 6 months [18]. Three other studies found a change in β-diversity [14,27,28] (one study in men only [14]), but these results were reported graphically and the direction of change could not be determined. No interventions reported increased β-diversity.

### 3.7. Changes in Relative Bacterial Abundance

Changes in relative bacterial abundance were assessed by 23 (82%) interventions. For the purpose of this review, we focused on changes at the taxonomic levels of phylum and genus only. Significant changes are shown in Table 2.

There were significant changes in six phyla across eight different interventions. Changes in the relative abundance of Bacteroidetes was inconsistent, increasing after a year-long fat-restricted diet and carbohydrate-restricted diet [7] and decreasing following a 16-week vegan diet [24]. Bacteroidetes also increased at 3 months of a low-fat diet and low-carbohydrate diet but returned to baseline levels by 12 months [18]. Three interventions reported a decrease in Firmicutes [7,15], with one reporting a decrease at 3 months, but not after 12 months [18]. The relative abundance of Proteobacteria decreased following two 16-week vegan diets [23,24] and a 3-month Mediterranean diet [27]. The ratio of Bacteroidetes to Firmicutes was unchanged in four interventions [23,24,25,27] and increased following a 12-week low-carbohydrate diet [25], but not reported in the majority of studies.

There were significant changes in 32 genera across 14 different interventions. The majority of genera only changed in one or two interventions, while changes in the other genera were inconsistent between studies. *Bifidobacterium* increased following two interventions (a normal-protein diet and a high-protein diet [16]) and decreased following two interventions (a high-protein, moderate-carbohydrate, non-ketogenic diet [17] and a low-carbohydrate, high-fat diet [21]). *Parabacteroides* increased following two interventions [27,28] and decreased in a third [26]. Another study found changes in *Bifidobacterium* and *Parabacteroides* abundance at 3 months, but these had returned to baseline levels by 12 months [18].

### 3.8. Changes in Faecal SCFAs

Seven interventions (25%) measured changes in faecal SCFA concentrations (Table 3). Concentrations were unaffected by all but one intervention in which butyrate concentration, but not total SCFA concentration, decreased [19].

### 3.9. Risk of Bias

RoB assessment for the 19 included studies is presented in Appendix A. Sequence generation was unclear in seven of the 14 trials that were randomised and only one trial adequately described the method of allocation concealment. RoB due to lack of blinding of participants/personnel and outcome assessors was deemed to be low in all studies, as lack of blinding is unlikely to affect microbiota-related outcomes or measurement of such. RoB due to incomplete outcome data was also rated as low in all studies as missing microbiota-related data are unlikely to be related to the true outcome. No studies had published protocols pre-specifying methods of microbiota analysis; as such, selective outcome reporting was unclear. All studies were deemed free from other sources of bias.

## 4. Discussion

The obese state has been associated with an altered gut microbiota, generating interest in the potential of weight loss to modulate the microbiota. This review found that dietary weight loss interventions had limited effect on bacterial diversity and faecal SCFA concentrations. Changes in bacterial abundance at the phylum and genus level were inconsistent across studies and there was no obvious correlation between macronutrient composition and microbiota outcomes.

The minimal effect of food-based weight loss interventions on α-diversity of the gut microbiota is consistent with other literature. A recent systematic review and meta-analysis of food-based, formula-based, and surgical weight loss interventions found a positive dose–response relationship between weight loss and α-diversity [30]. Food-based dietary interventions on their own, however, had an inconsistent effect on α-diversity, with no statistically significant effect when results were pooled [30]. It is likely that the degree of weight loss achieved through food-based weight loss is not large enough to produce the statistically significant change in α-diversity seen within very-low-calorie formula-based diets (VLCD) and with surgical interventions, both of which were excluded from this review. 

In addition, microbiota metrics were not typically the primary outcome in the studies included in this review. The weight loss interventions reported power calculations based on detecting significance in weight loss rather than microbiota changes. Given the large interindividual variability in the gut microbiota, much larger sample sizes would be needed to detect significant changes following diet-induced weight loss. 

Low fibre intake in the included studies may also explain the lack of consistent effect on the gut microbiota. In studies that reported fibre intake, this ranged from 10 to 33 g per day. Fibre is the main substrate for bacterial fermentation and observational studies of rural African tribes indicate that high-fibre diets are associated with greater bacterial diversity and SCFA production [31,32]. These tribes consume upwards of 100 g of fibre per day and similarly high levels may be needed to induce microbiota changes in interventional studies, which is unlikely to be feasible without the use of supplements. The diversity of plant foods consumed is also important, with the American Gut Project finding microbial diversity to be associated with the number of unique plant foods consumed each week rather than self-reported categories such as “vegan” or “omnivore” [33]. Low dietary diversity may explain the unchanged or decreased bacterial diversity seen in the two vegan dietary interventions included in this review. Richer and more robust dietary reporting methods, including details on soluble/insoluble fibre intake as well as the type and diversity of plant foods consumed, are needed to better understand the relationship between diet and the microbiota [34]. 

The finding that dietary weight loss strategies have a limited effect on microbiota-related outcomes is surprising considering previous research showing that 4 days of a completely animal- or plant-based diet rapidly alters gut microbial communities [35]. This suggests that drastic dietary changes are needed to observe an effect. It may also be that the microbiota is resistant to long-term changes [36]. Indeed, a study included in this review observed changes in relative bacterial abundance at 3 months, but these were ameliorated by 6 months despite continued dietary adherence [18]. Long-term studies with frequent microbiota measurements are required to examine the resilience of the microbiota to dietary changes. Differences in baseline microbiota characteristics may also explain inconsistencies across studies, with baseline microbial diversity and gene richness associated with individualized gut microbiota responses [37,38,39]. 

Gut bacteria produce a wide range of metabolites that have been implicated in health outcomes [40]. SCFAs are among the most commonly measured metabolites in microbiome studies; as such, we limited our review to these metabolites only. Only four studies (21%) meeting inclusion criteria analysed faecal SCFA concentrations, which were mostly unchanged following dietary intervention. Further studies assessing the effect of food-based weight loss interventions on SCFAs, as well as other microbiota-derived metabolites such as trimethylamine N-oxide, secondary bile acids, and tryptophan metabolites, are needed to facilitate a meta-analysis. 

Differences in study design, population, and methodology limit the conclusions that can be drawn from this review. While we aimed to exclude studies involving participants with comorbidities such as metabolic syndrome, presence of comorbidities was not always described in the included studies. Differences in age, sex, geographic location, and inclusion/exclusion criteria may also represent confounding factors. Several different molecular biology techniques (16S rRNA amplicon sequencing, shotgun metagenomic sequencing, FISH, qPCR) were used to assess the gut microbial composition. Differences in the 16S rRNA gene region amplified and OTU picking protocols and databases may also explain differing results [41]. A wide range of metrics were used to assess α-diversity (e.g., Shannon index, Pielou index, Chao1 index) and β-diversity (e.g., Bray–Curtis dissimilarity, Aitchison distance, weighted and unweighted UniFrac distances), making a meta-analysis not possible. Many studies reported relative changes (e.g., an increase in diversity or decrease in a particular taxa) rather than absolute value changes, further limiting our ability to conduct a meta-analysis. Reporting absolute percentage changes in relative bacterial abundance, as per Kahleova et al. [24], would facilitate quantitative comparison between studies. Due to the lack of species-level sensitivity of 16S rRNA-based techniques, we were only able to compare changes at the genus level. Further research utilising whole-genome sequencing is needed to evaluate the effect of dietary interventions on individual species. Reporting of baseline dietary intake and microbiota composition is also needed to evaluate whether changes are observed only in participants who drastically alter their diet or in those with low bacterial gene richness or lacking certain taxa to begin with.

## 5. Conclusions

There were minimal changes in bacterial diversity and faecal SCFA concentrations following dietary weight loss interventions, with inconsistent changes in relative bacterial abundance at the phylum and genus level. Further studies, adequately powered to detect changes in microbiota-related outcomes, are needed. Greater consistency in the method of microbiota analysis and α- and β-diversity metrics, as well as reporting of absolute changes in these variables, is needed if a meta-analysis is to be conducted.

## Figures and Tables

**Figure 1 nutrients-14-01953-f001:**
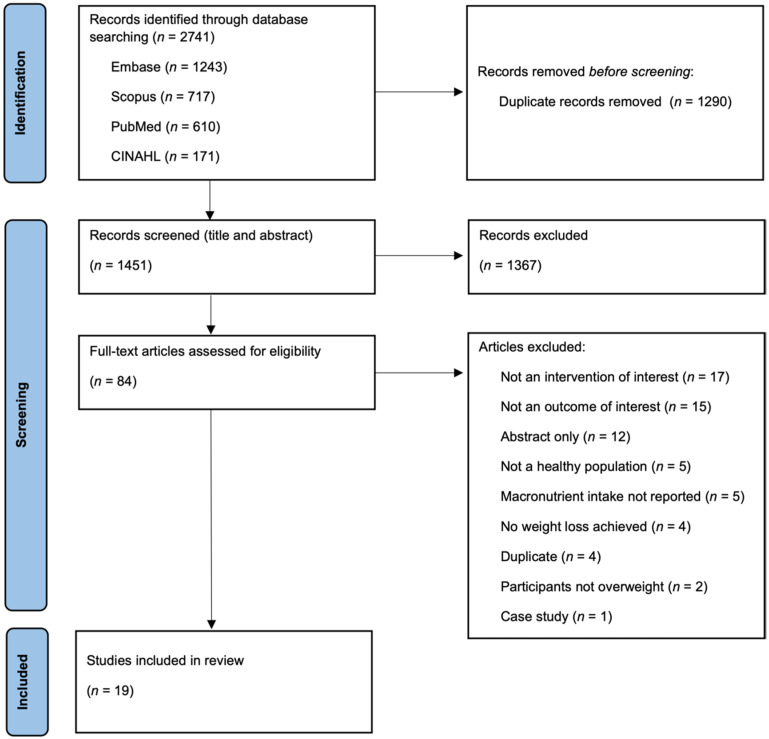
Flow diagram for study selection.

**Table 1 nutrients-14-01953-t001:** Characteristics of included studies.

Trial, Country	*n * (Female)	Age, Years	BMI, kg/m^2^	Intervention Protocol	Duration	Microbiota Analysis Method
Bendsten 2018, Denmark [12]	40 (35)	42 (1)	31.5 (0.4)	High-dairy diet: 18% P, 52% C, 30% F, 500 kcal/day deficit, 1500 mg calcium/day	24 weeks	16S rRNA (V3–V4)
	40 (34)	45 (2)	30.8 (0.4)	Low-dairy diet: 18% P, 52% C, 30% F, 500 kcal/day deficit, 600 mg calcium/day	24 weeks	16S rRNA (V3–V4)
Benítez-Páez 2021, Denmark [13]	59 (39)	48.9 (8.6)	32.8 (3.9)	Calorie-restricted diet + fibre: 18–20% P, 52–53% C, 32–33% F, 500 kcal/day deficit, 14–22 g/day fibre + 20 g/day prebiotic fibre supplement (10 g inulin + 10 g resistant maltodextrin)	12 weeks	Shotgun metagenomics
	57 (37)	48.4 (8.3)	34.4 (4.4)	Calorie-restricted diet + placebo: 18–20% P, 52–53% C, 32–33% F, 500 kcal/day deficit, 14–22 g/day fibre + placebo supplement (maltodextrin)	12 weeks	Shotgun metagenomics
Cuevas-Sierra 2021, Spain [14]	82 (54)	NR	M: 31.9 (3.2) F: 30.9 (3.2)	Moderately high-protein diet: 30% P, 40% C, 30% F, 30% energy restriction	16 weeks	16S rRNA (V3–V4)
	97 (70)	NR	M: 32.1 (3.5) F: 31.9 (3.9)	Low-fat diet: 18% P, 60% C, 22% F, 30% energy restriction	16 weeks	16S rRNA (V3–V4)
Dhakal 2020, USA [15]	58 (44)	45.7 (15.8)	34.6 (7.2)	Retail weight reduction program	12 weeks	16S rRNA (V4)
Dong 2020, USA [16]	43 (10)	55.9 (10.1)	34.9 (4.5)	High-protein diet: 30% P, 40% C, 30% F, 2 weeks ad libitum then 6 weeks 500 kcal/day deficit	8 weeks	16S rRNA (V4)
	37 (8)	55.7 (11.4)	34.6 (5.1)	Normal-protein diet: 15% P, 55% C, 30% F, 2 weeks ad libitum then 6 weeks 500 kcal/day deficit	8 weeks	16S rRNA (V4)
Duncan 2008, Scotland [17]	23 (0)	NR	>30	High-protein, moderate-carbohydrate, non-ketogenic diet: 30% P, 35% C, 35% F, <8.5 MJ/day, 164 g/d CHO, 12.2 g/day non-starch polysaccharide	4 weeks	16S rRNA-based quantitative FISH
Fragiadakis 2020, USA [18]	25 (20)	42.6 (5.8)	32.8 (3.9)	Low-carbohydrate diet	12 months	16S rRNA (V4)
	24 (19)	39.2 (5.5)	33.7 (3.5)	Low-fat diet	12 months	16S rRNA (V4)
Gratz 2019, Scotland [19]	18 (0)	49 (12)	36.6 (5.8)	Participants followed a 7-day weight maintenance diet followed by three 10-day weight loss diets in a randomized crossover design without washout: 1. Normal-protein diet: 15% P, 55% C, 30% F, energy = 1 × BMR 2. Normal-protein diet enriched with free amino acids and moderate amounts of carbohydrate: 15% P, 15% free amino acids, 40% C, 30% F, energy = 1 × BMR 3. High-protein diet containing moderate amounts of carbohydrate: 30% P, 40% C, 30% F, energy = 1 × BMR	37 days	None
Gutiérrez-Repiso 2021, Spain [20]	21 (11)	64.0 (4.7)	33.4 (3.3)	Mediterranean diet: 20% P, 40–45% C, 35–40% F, 8–10% SFA, 600 kcal/day deficit, 150 min/week walking	6 months	16S rRNA (NR)
Jaagura 2021, Estonia [21]	27 (NR)	NR	28.9–44.4	Low-carbohydrate, high-fat weight loss diet: 30 ± 10% energy deficit	4 weeks	16S rRNA (V3–V4)
Johnstone 2020, UK [22]	24 (16)	20–62	32.8 (4.07)	Weight loss diet: 30% P, 40% C, 30% F, 25 g/day fibre, energy intake = RMR	3 weeks	qPCR, 16S rRNA (NR)
Kahleova 2020, USA [23]	84 (69)	52.9 (11.7)	32.6 (3.7)	Low-fat vegan diet: 20–30 g/day fat, high in vegetables, grains, legumes, and fruit, instructed to avoid animal products and added oil, vitamin B12 supplemented (500 μg/day)	16 weeks	16S rRNA (V4)
Kahleova 2021, USA [24]	62 (48)	57.4 (NR)	34.0 (NR)	Low-fat vegan diet: consisted of fruits, vegetables, grains, and legumes. Animal products and added fats were excluded. Vitamin B12 was supplemented (500 μg/day)	16 weeks	16S rRNA (V4)
Ley 2006, USA [7]	6 (4)	53.7 (NR)	>30	Fat-restricted diet: 30% F, 10–15 g/day fibre, 1200–1500 kcal/day for women, 1500–1800 kcal/day for men	12 months	16S rRNA (NR)
	6 (5)	42.0 (NR)	>30	Carbohydrate-restricted diet: 25% C, 10–15 g/d fibre, 1200–1500 kcal/day for women, 1500–1800 kcal/day for men	12 months	16S rRNA (NR)
Ma 2021, China [25]	25 (25)	NR	26.6 (0.5)	Low-carbohydrate diet: 20 g/day carbohydrates in the first week, then 10 g/day extra weekly until reaching 120 g/day at the end of the intervention	12 weeks	Shotgun metagenomics
	25 (25)	NR	26.9 (0.4)	Calorie-restricted diet: 1200 kcal/day, 20% P, 55% C, 25% F, 10% SFA, 300 mg/day cholesterol	12 weeks	Shotgun metagenomics
Nogacka 2021, Spain [26]	9 (4)	49.67 (7.81)	>40	Hypocaloric diet: 15% P, 55% C, 30% F, <10% SFA, 20–25 g/day fibre, 20 kcal/kg body weight (~1800–2000 kcal/day)	6–8 months	16S rRNA (NR)
Pisanu 2020, Italy [27]	23 (20)	53 (9)	35.2 (4.3)	Mediterranean diet: 20% P, 55% C, 25% F, ≥25 g/day fibre, energy = BMR (±10%)	3 months	16S rRNA (V3–V4)
Stanislawksi 2021, USA [28]	71 (NR)	40.7 (9.8)	33.1 (4.4)	Energy-restricted diet: 34% weekly energy deficit achieved through either daily caloric restriction or intermittent fasting (80% energy deficit on 3 non-consecutive days each week). Moderate intensity physical activity: 300 min per week.	12 weeks	16S rRNA (V3–V4)
Zhang 2021, China [29]	26 (22)	36.58 (8.70)	30.44 (3.38)	Low-carbohydrate diet: 10–25% C, no energy restriction	12 weeks	16S rRNA (V3–V4)

% *p*: percent of energy from protein, % C: percent of energy from carbohydrates, % F: percent of energy from fat, M: males, F: females, NR: not reported, FISH: fluorescence in situ hybridization. Age and BMI reported as mean (SD).

**Table 2 nutrients-14-01953-t002:** Summary of the microbiota changes of included studies.

Trial	Reported Dietary Intake	Weight Loss, kg	α-Diversity	β-Diversity	Relative Bacterial Abundance
Bendsten 2018 [12]	High-dairy diet: 1649 kcal, 21% P, 47% C, 31% F, 20 g fibre	6.6 (1.3)	↔ Shannon	↔ UniFrac	↔
	Low-dairy diet: 1585 kcal, 19% P, 46% C, 32% F, 22 g fibre	7.9 (1.5)	↔ Shannon	↔ UniFrac	↓ *Veillonella*
Benítez-Páez 2021 [13]	Calorie-restricted diet + fibre: 1642 kcal, 21% P, 47% C, 31% F, 18 g fibre	6.1 (NR)	↔ Simpson	↔ B–C	NR
	Calorie-restricted diet + placebo: 1730 kcal, 21% P, 46% C, 32% F, 18 g fibre	5.5 (NR)	↔ Simpson	↔ B–C	NR
Cuevas-Sierra 2021 [14]	Moderately high-protein diet: M: 33% P, 50% C, 17% F: 34% P, 49% C, 17% F	M: 10.3 (NR) F: 8.9 (NR)	M: ↔ Shannon F: ↔ Shannon	M: ↔ B–C F: ↔ B–C	↑ *Granulicatella* ↓ *Phascolarctobacterium, Dielma*
	Low-fat diet: M: 25% P, 61% C, 14% FF: 24% P, 63% C, 13% F	M: 11.0 (NR) F: 8.6 (NR)	M: ↑ Shannon F: ↔ Shannon	M: ↑↓ B–C F: ↔ B–C	↔
Dhakal 2020 [15]	Retail weight reduction program: 1818 kcal, 24% P, 38% C, 38% F, 18 g fibre	10.2 (NR)	↑ OTU richness ↔ Shannon	NR	↑ Tenericutes, Euryarchaeota ↓ *Firmicutes*, *p_Actinobacteria*
Dong 2020 [16]	High-protein diet: NR	3.5 (NR)	↑ Shannon	↔ Aitchison	↑ *Akkermansia*, *Bifidobacterium* ↓ *Prevotella_9*
	Normal-protein diet: NR	2.8 (NR)	↔ Shannon	↔ Aitchison	↑ *Akkermansia*, *Bifidobacterium* ↓ *Prevotella_9*
Duncan 2008 [17]	High-protein, moderate-carbohydrate, non-ketogenic diet: NR	4.6 (NR)	NR	NR	↑ *Clostridium coccoides*-related bacteria (other than *Roseburia* + *Eubacterium rectale*) ↓ Total bacterial number, *Roseburia* + *Eubacterium rectale*, *Bifidobacterium*
Fragiadakis 2020 [18]	Low-carbohydrate diet: 426 kcal/d deficit, 22% P, 32% C, 43% F, 18 g fibre	5.1 (6.7)	↔ Observed ASVs	3 months: ↓ B–C 6 months: ↔ B–C 12 months: ↔ B–C	3 m: ↑ Bacteroidetes, *Bacteroides*, *Parabacteroides, Sutterella, Bilophila, Desulfovibrio, Butyricimonas, Lachnospira, Oscillospira* 12 m: ↔
Fragiadakis 2020 [18]	Low-fat diet: 484 kcal/d deficit, 21% P, 48% C, 29% F, 20 g fibre	5.6 (5.7)	↔ Observed ASVs	3 months: ↔ B–C 6 months: ↔ B–C 12 months: ↔ B–C	3 m: ↑ Bacteroidetes, *Bacteroides, Parabacteroides* 3 m: ↓ Actinobacteria, Firmicutes, *Bifidobacterium, Dorea, Blautia, Ruminococcus* 12 m: ↔
Gutiérrez- Repiso 2021 [20]	Mediterranean diet: NR	7.8 (1.9)	↑ Observed ASVs ↑ Shannon ↑ Faith ↑ Pielou	↔ UniFrac	↑ *Faecalibacterium*
Jaagura 2021 [21]	Low-carbohydrate, high-fat weight loss diet: 25% P, 23% C, 50% F, 12 g fibre/1000 kcal	7.7 (2.5)	↔ Observed species ↔ Shannon	↓ B–C	↑ *Alistipes, Butyricimonas, Odoribacter, Ruminococcus_1* ↓ *Bifidobacterium, Collinsella, Dorea*
Johnstone 2020 [22]	Weight loss diet: 1930 kcal, 29% P, 40% C, 30% F, 10% SFA, 25 g fibre, 15 g insoluble fibre, 5 g soluble fibre, 7 g resistant starch	2.8 (NR)	↔ Chao1 ↔ Shannon	NR	↔
Kahleova 2020 [23]	Low-fat vegan diet: 1294 kcal, 43 g P (13%), 236 g C (73%), 24.3 g F (17%), 33 g fibre, 9 g soluble fibre, 25 g insoluble fibre	6.4 (NR)	↔ AWPD	NR	↑ *Faecalibacterium* ↓ Proteobacteria ↔ Bacteroidetes:Firmicutes, butyrate producing bacteria
Kahleova 2021 [24]	Low-fat vegan diet: 1315 kcal, 12% P, 69% C, 17% F, 33 g fibre, 9 g soluble fibre, 24 g insoluble fibre	6.0 (NR)	↓ AWPD	NR	↑ *Eubacterium* ↓ Bacteroidetes, Proteobacteria ↔ Bacteroidetes:Firmicutes, butyrate-producing bacteria
Ley 2006 [7]	Fat-restricted diet: NR	15.4 (NR)	↔ Shannon	NR	↑ Bacteroidetes ↓ Firmicutes
	Carbohydrate-restricted diet: NR	8.0 (NR)	↔ Shannon	NR	↑ Bacteroidetes ↓ Firmicutes
Ma 2021 [25]	Low-carbohydrate diet: 1195 kcal, 26% P, 36% C, 38% F, 10 g fibre	5.3 (NR)	↔ Shannon	↓ B–C	↑ Bacteroidetes:Firmicutes
	Calorie-restricted diet: 1355 kcal, 18% P, 51% C, 31% F, 11 g fibre	5.1 (NR)	↔ Shannon	↔ B–C	↔ Bacteroidetes:Firmicutes
Nogacka 2021 [26]	Hypocaloric diet: NR	Group 1: <5% BW (*n* = 5) Group 2: >5% BW (*n* = 4)	Group 2 vs. total at baseline: ↔ Chao1 ↔ Shannon	NR	Group 2 vs. total at baseline: ↑ *Clostridum sensu stricto 1* ↓ *Parabacteroides*
Pisanu 2020 [27]	Mediterranean diet: 1341 kcal, 19% P, 50% C, 29% F, 17 g fibre	6.7 (NR)	↔ Shannon	↑↓ B–C	↑ *Catenibacterium, Caldilinea, Parabacteroides, Sphingobacterium, Veillonella* ↓ Proteobacteria, *Megamonas, Roseburia, Ruminococcus, Streptococcus, Sutterella* ↔ Bacteroidetes:Firmicutes
Stanislawksi 2021 [28]	Energy-restricted diet: 1276 kcal, 21% P, 42% C, 35% F	5.8 (3.8)	↑ Observed OTUs ↑ Evenness ↑ Shannon ↑ Faith	↑↓ UniFrac	↑ *Parabacteroides, Alistipes, Bacteroides* ↓ *Subdoligranulum, Collinsella*
Zhang 2021 [29]	Low-carbohydrate diet: 1470 kcal, 34% P, 16% C, 50% F	2.2 (1.2) kg/m^2^	↔ Shannon ↔ Simpson ↔ Richness (genus level)	↔ B–C	↔ (phylum level)

% *p*: percent of energy from protein, % C: percent of energy from carbohydrates, % F: percent of energy from fat, M: males, F: females, BW: body weight, NR: not reported, ↑: increase, ↓: decrease, ↔: no change, ↑↓: direction of change not reported, AWPD: abundance-weighted phylogenetic diversity measure, B–C: Bray–Curtis dissimilarity. Weight loss reported as mean (SD).

**Table 3 nutrients-14-01953-t003:** Changes in faecal short-chain fatty acid concentrations.

Trial	Reported Dietary Intake	Weight Loss, kg	Total SCFAs	Butyrate	Propionate	Acetate
Benítez-Páez 2021 [13]	Calorie-restricted diet + fibre: 1642 kcal, 21% P, 47% C, 31% F, 18 g fibre	6.1 (NR)	NR	↔	↔	↔
	Calorie-restricted diet + placebo: 1730 kcal, 21% P, 46% C, 32% F, 18 g fibre	5.5 (NR)	NR	↔	↔	↔
Gratz 2019 [19]	Normal-protein weight loss diet: 2154 kcal, 80 g P (15%), 309 g C (57%), 73 g F (31%), 29 g fibre	3.9 (NR)	↔	↔	↔	↔
	Normal-protein weight loss diet enriched with free amino acids and moderate amounts of carbohydrate: 2143 kcal, 156 g P (29%), 219 g C (41%), 73 g F (31%), 20 g fibre	4.3 (NR)	↔	↔	↔	↔
	High-protein weight loss diet containing moderate amounts of carbohydrate: 2106 kcal, 153 g P (29%), 219 g C (42%), 72 g F (31%), 18 g fibre	4.0 (NR)	↔	↓	↔	↔
Johnstone 2020 [22]	Weight loss diet: 1930 kcal, 29% P, 40% C, 30% F, 10% SFA, 25 g fibre, 15 g insoluble fibre, 5 g soluble fibre, 7 g resistant starch	2.8 (NR)	NR	↔ (% of total SCFA)	↔ (% of total SCFA)	↔ (% of total SCFA)
Nogacka 2021 [26]	Hypocaloric diet: NR	Group 1: <5% BW (*n* = 5) Group 2: >5% BW (*n* = 4)	↔ (Group 2 vs. total at baseline)	↔ (Group 2 vs. total at baseline)	↔ (Group 2 vs. total at baseline)	↔ (Group 2 vs. total at baseline)

% *p*: percent of energy from protein, % C: percent of energy from carbohydrates, % F: percent of energy from fat, NR: not reported, ↓: decrease, ↔: no change.

## Data Availability

Not applicable.

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
