# Peer review of "Impact of Food-Based Weight Loss Interventions on Gut Microbiome in Individuals with Obesity: A Systematic Review"

_nutrients, 2022, doi:10.3390/nu14091953_

Round 1
Reviewer 1 Report
The proposed manuscript from Bliesner et al. is a short, easy to read, systematic review without meta-analysis on the effect of food-based weight loss diet on the gut microbiota. Even if another SR with meta-analysis (ref 30) was published shortly on a similar but broader theme, this manuscript can be interesting as considering several studies published recently, which were not taken into account in the SR with MA.
I however have some recommendation to improve it.
First, I find the title misleading. In the abstract it becomes clear that formula diet studies are excluded but not in the title. Please reformulate the sentence, so that it is clear already in the title.
Materials and Methods:
-search strategy: I understand the search criteria NOT “child” but why NOT “adult”?
-informatic tools used should have a version number or so (Covidence, RoB). I was also wondering why RoB and not RoB2 was used, as the last one was released already 2019.
Results:
-if the exclusion criteria mentioned in Fig1 and suppl. Table S2 are understandable, I would like to have more details about two of them: not an intervention of interest (please mention shortly in Table S2 what then the intervention was: obviously not food-based but formula-based or something else?), and not an outcome of interest (no real microbiota data or what?).
-microbiota: It would be interesting to mention why the chosen microbial parameters were selected. If a high a-diversity is rather associated with a healthy microbiota and could be a useful “target” to improve in obesity-associated dysbiosis, there are no such meaning behind b-diversity. This one is important to have an overview of the whole impact of an intervention on the microbiota and is therefore mostly shown as a PCoA or NMDS plot with different dots for different time points or group. It is showing if strong changes occurred or not. Then one has to look at the details, what exactly was changed, ie which taxa. They mentioned 3 papers showing a decrease after intervention. In one of them (ref 13), I could not find any such decrease. In ref 21, there is indeed a decrease in dissimilarity in both groups (fig A3), which would indicate a trend to a more homogeneous group (microbiotas of different persons are less different after intervention than before). This could be due to more homogenous diet but is not so easy to interpret. I unluckily do not have access to the third reference mentioned. Try to better explain your intention looking at b-div in this review. About table2, it would be informative to give the metrics used for a-div (and b-div, if kept) in each study (like for ref 15), as it would allow the reader to better assess the results.
Also why looking only at SCFA? Sure, they are the most analysed microbiota metabolites, but other metabolites might be relevant for health (within or outside obesity context). At least a short explanation on this is needed.
Discussion: well written but a bit short. Please add more citations (see below)
L232: you mention a correlation between macronutrient composition and microbiota parameters: did you check for this? Can you extract from the selected studies enough macronutrient information to perform any statistics? This would be really interesting.
L263: do you know Johnson et al 2019, Cell Host & microbe 25, 789-802? Shows nicely that nutrient profiles are not so useful to explain microbiome as single food (particularly plant)… would fit as reference here…
L267: any references (not included in the review) to sustain both claims: drastic dietary changes and resilience in long term…? Also the fact that some microbiota are more or less permissive to changes (low gene richness) should be discussed at least shortly here.
L279: indeed 16S-microbiota analysis are always biased, through the choice of the V-region, primer sequences, PCR type, sequencer, bioinformatic pipeline and used databases. Some references on these aspects are necessary. As some of these have a stronger impact than others, it would be informative to mention, let’s say, at least the V-region analysed in each study (in table1). This would help the reader to better assess the comparability of the mentioned studies.
L284: indeed lack of absolute quantification is a limiting factor for comparability between studies. Some references on this would also be welcome here.
Conclusion:
Authors claim that no meta-analysis is possible. But ref. 30 managed it, also in subgroups. As most study used 16S rRNA gene sequencing and at least one metric for alpha-diversity, it should be possible to perform some analysis on all those with the same metric. Adding this info in table2 would allow a better understanding (if indeed not possible).
Author Response
The proposed manuscript from Bliesner et al. is a short, easy to read, systematic review without meta-analysis on the effect of food-based weight loss diet on the gut microbiota. Even if another SR with meta-analysis (ref 30) was published shortly on a similar but broader theme, this manuscript can be interesting as considering several studies published recently, which were not taken into account in the SR with MA.
I however have some recommendation to improve it.
First, I find the title misleading. In the abstract it becomes clear that formula diet studies are excluded but not in the title. Please reformulate the sentence, so that it is clear already in the title.
Thank you for this comment. The title has been revised accordingly to make the intervention of interest clearer.
Materials and Methods:
-search strategy: I understand the search criteria NOT “child” but why NOT “adult”?
The search term NOT "child" excludes papers where both adults and children were included in the study - we may potentially be able to extract the adult data from these papers. The search term "child" NOT "adult" returns results only in children, excluding papers that include both children and adults. Hence, the search term NOT("child" NOT "adult") returns studies exclusively in adults as well as studies with a mix of adults and children from which we can extract the adult data.
-informatic tools used should have a version number or so (Covidence, RoB). I was also wondering why RoB and not RoB2 was used, as the last one was released already 2019.
This is a valid comment. The materials and methods have been updated to include a rationale on the use of the older version. Version numbers have been added also; thank you for picking this up.
Results:
-if the exclusion criteria mentioned in Fig1 and suppl. Table S2 are understandable, I would like to have more details about two of them: not an intervention of interest (please mention shortly in Table S2 what then the intervention was: obviously not food-based but formula-based or something else?), and not an outcome of interest (no real microbiota data or what?).
A good suggestion. Table S2 has been updated to include greater details on exclusion reasons.
-microbiota: It would be interesting to mention why the chosen microbial parameters were selected. If a high a-diversity is rather associated with a healthy microbiota and could be a useful “target” to improve in obesity-associated dysbiosis, there are no such meaning behind b-diversity. This one is important to have an overview of the whole impact of an intervention on the microbiota and is therefore mostly shown as a PCoA or NMDS plot with different dots for different time points or group. It is showing if strong changes occurred or not. Then one has to look at the details, what exactly was changed, ie which taxa. They mentioned 3 papers showing a decrease after intervention. In one of them (ref 13), I could not find any such decrease. In ref 21, there is indeed a decrease in dissimilarity in both groups (fig A3), which would indicate a trend to a more homogeneous group (microbiotas of different persons are less different after intervention than before). This could be due to more homogenous diet but is not so easy to interpret. I unluckily do not have access to the third reference mentioned. Try to better explain your intention looking at b-div in this review. About table2, it would be informative to give the metrics used for a-div (and b-div, if kept) in each study (like for ref 15), as it would allow the reader to better assess the results. Also why looking only at SCFA? Sure, they are the most analysed microbiota metabolites, but other metabolites might be relevant for health (within or outside obesity context). At least a short explanation on this is needed.
Thank you for your comment. Β-diversity was included in this review to evaluate whether the dietary interventions produced an overall change in the gut microbiome, i.e. to assess whether they were effective or not. Upon review, there was indeed no change in β-diversity in ref 13; the table and text have been updated accordingly. Thank you for picking up this error. We agree that including measures of α- and β-diversity would be valuable to the reader and have updated Table 2 to include this information. The choice to focus on SCFAs only is a valid concern. SCFAs were the most commonly reported metabolite across the included studies; few other metabolites were measured and reported on. A short explanation on this has been added to the discussion.
Discussion: well written but a bit short. Please add more citations (see below)
L232: you mention a correlation between macronutrient composition and microbiota parameters: did you check for this? Can you extract from the selected studies enough macronutrient information to perform any statistics? This would be really interesting.
This would indeed be interesting. However, based on the lack of obvious association between diet type and microbiota outcomes, we felt a statistical analysis was unnecessary. Additionally, five of the included studies did not measure or report on the actual macronutrient intake of study participants, limiting the data available for analysis.
L263: do you know Johnson et al 2019, Cell Host & microbe 25, 789-802? Shows nicely that nutrient profiles are not so useful to explain microbiome as single food (particularly plant)… would fit as reference here…
Thank you for this excellent reference; it has been added to the review.
L267: any references (not included in the review) to sustain both claims: drastic dietary changes and resilience in long term…?
Ref 34 suggests that drastic dietary changes may be required; the text has been updated to make this clearer. A reference has been added to substantiate the possibility of long-term resilience of the microbiome.
Also the fact that some microbiota are more or less permissive to changes (low gene richness) should be discussed at least shortly here.
A good point. A brief discussion of this has been added with references.
L279: indeed 16S-microbiota analysis are always biased, through the choice of the V-region, primer sequences, PCR type, sequencer, bioinformatic pipeline and used databases. Some references on these aspects are necessary.
A valid comment. References on these aspects have been added.
As some of these have a stronger impact than others, it would be informative to mention, let’s say, at least the V-region analysed in each study (in table1). This would help the reader to better assess the comparability of the mentioned studies.
This would indeed be useful for the reader. V-regions analysed have been added to Table 1.
L284: indeed lack of absolute quantification is a limiting factor for comparability between studies. Some references on this would also be welcome here.
While no reference could be found on this per se, we have referenced a study from this review reporting on absolute changes in relative bacterial abundance pre- vs post-intervention for the reader's interest.
Conclusion:
Authors claim that no meta-analysis is possible. But ref. 30 managed it, also in subgroups. As most study used 16S rRNA gene sequencing and at least one metric for alpha-diversity, it should be possible to perform some analysis on all those with the same metric. Adding this info in table2 would allow a better understanding (if indeed not possible).
We agree that a meta-analysis would have been desirable. However, the meta-analysis you reference pooled different measures of α-diversity, which we did not think was appropriate for our purposes given that these measures represent different aspects of diversity and therefore cannot be directly compared. While a number of studies used the Shannon index, most did not report this in a way that was conducive to meta-analysis, i.e. they did not report absolute values pre- and post-intervention.
Reviewer 2 Report
Thank you for an excellent paper. I totally agree with your conclusions "There were minimal changes in bacterial diversity and faecal SCFA concentrations following dietary weight loss interventions, with inconsistent changes in relative bacterial abundance at the phylum and genus level".
Author Response
Reviewer 2
Thank you for an excellent paper. I totally agree with your conclusions "There were minimal changes in bacterial diversity and faecal SCFA concentrations following dietary weight loss interventions, with inconsistent changes in relative bacterial abundance at the phylum and genus level".
Thank you for reviewing this paper and your kind comment.